# The Role of Protein Ubiquitination in the Onset and Progression of Sepsis

**DOI:** 10.3390/cells14131012

**Published:** 2025-07-02

**Authors:** Meng-Yan Chen, Yang Liu, Min Fang

**Affiliations:** 1School of Life Sciences, Henan University, Kaifeng 475004, China; cmyan@henu.edu.cn (M.-Y.C.); liuyang1@henu.edu.cn (Y.L.); 2Henan Key Laboratory of Synthetic Biology and Biomanufacturing, Henan University, Kaifeng 475004, China

**Keywords:** ubiquitination, sepsis, NF-κB, NLRP3 inflammasome, septic organ injury, macrophage polarization, cardiomyopathy

## Abstract

Sepsis is a life-threatening condition characterized by a dysregulated host response to infection, with complex pathophysiological mechanisms. As an important post-translational modification, protein ubiquitination exhibits multiple non-traditional functions in sepsis beyond its conventional role in protein degradation. Regulating the network of inflammatory cytokines, the dynamic balance of immune cells and organ-specific protective pathways is deeply involved in the pathological process of sepsis. This review focuses on the unconventional roles of protein ubiquitination in sepsis, including its regulation of the inflammatory response, immune cell functions, and organ protection. It systematically summarizes the regulatory mechanisms of ubiquitination in the non-degradative activation of the nuclear factor kappa B (NF-κB) signaling pathway, the dynamic assembly of the NLRP3 inflammasome, the reprogramming of macrophage polarization, and the injuries of organs such as the heart, liver, and lungs. These processes demonstrate that ubiquitination serves as a pivotal nexus between immunological dysregulation and multi-organ impairment in sepsis. This review suggests that targeting non-degradative ubiquitination alterations may provide viable therapeutic options to mitigate excessive inflammation and organ failure in sepsis.

## 1. Introduction

Sepsis is a life-threatening condition characterized by a dysregulated host response to infection leading to organ dysfunction [1,2]. As a severe challenge in the global public health field, sepsis has always been one of the focuses of medical research. According to the global epidemiological data from The Lancet in 2020, approximately 50 million people worldwide suffer from sepsis each year [3,4,5]. The mortality rate is as high as 20% [5]. Its core pathological feature is the dysregulation of the host immune response triggered by infection, manifested as a dynamic imbalance between early excessive inflammation, such as the tumor necrosis factor-alpha (TNF-α) and interleukin-6 (IL-6) storms, and late immunosuppression, such as T cell exhaustion, ultimately leading to multiple organ dysfunction syndrome (MODS) [6]. Organ dysfunction caused by sepsis is systemic and leads to the impairment of organs far from the primary infection site, including the heart, lungs, liver, and kidneys [7,8]. Among the affected organs, sepsis-induced myocardial dysfunction (SIMD) is particularly critical [9,10]. It is manifested as acute cardiac dysfunction, with an incidence rate of 13.8–40% and a mortality rate as high as 70–90%, posing a serious threat to human health [11,12]. Although its pathological mechanism has not been fully elucidated, recent studies have shown that protein ubiquitination, a non-traditional post-translational modification (PTM), plays a central regulatory role in immune imbalance and organ injury in sepsis [13].

Protein ubiquitination involves an E1-E2-E3 enzyme cascade reaction [13]. This process attaches ubiquitin molecules (76 amino acids) to target proteins via isopeptide bonds at lysine residues or unconventional bonds at the N-terminus or cysteine to form various chain types [13,14]. For example, K48 chains mediate proteasomal degradation, K63 chains facilitate signal activation, and linear M1 chains regulate inflammation [15]. Unlike the traditionally recognized “proteasomal degradation function”, recent studies have revealed that in sepsis, protein ubiquitination regulates the inflammatory signaling pathway, immune cell polarization, and organ protection mechanisms through non-degradative modifications such as the K63/M1 chains [16,17]. For example, the linear ubiquitin assembly complex (LUBAC) modifies receptor-interacting protein kinase 1 (RIPK1) by generating M1 chains, and the deubiquitinating enzyme ubiquitin-specific peptidase 5 (USP5) removes the K63-linked polyubiquitin chains from RIPK1 to inhibit its activity, thereby suppressing the necroptosis pathway to protect cardiomyocytes [17,18]. The E3 ligase tripartite motif-containing protein 27 (TRIM27) exacerbates oxidative stress in lung tissues by degrading peroxisome proliferator-activated receptor γ (PPARγ) through K48 ubiquitination [19]. These findings provide a new direction for the intervention of the “ubiquitination axis” in the treatment of sepsis. This review synthesizes recent evidence to dissect the non-traditional roles of protein ubiquitination in sepsis, focusing on its regulatory mechanisms in inflammatory pathway activation, such as nuclear factor kappa B (NF-κB) non-degradative signaling, in immune cell phenotypic plasticity such as macrophage M1/M2 polarization, and in organ-specific protective responses such as liver anti-oxidative stress pathways. By integrating molecular mechanisms with clinical implications, we aim to establish ubiquitination as a novel therapeutic target for sepsis, bridging basic research with translational medicine.

## 2. Mechanisms of Ubiquitination

Ubiquitination is a PTM in which one, or usually more, chains of 76 amino acids are covalently linked to lysine residues within the substrate proteins [20]. Ubiquitination can affect the functions of proteins in various ways, such as influencing protein stability, turnover, and cellular localization, and inducing conformational changes to affect interactions with other proteins [21]. This process requires the sequential participation of E1 ubiquitin-activating enzymes, E2 ubiquitin-conjugating enzymes, and E3 ubiquitin ligases (Figure 1). There are three main types of E3 ligases: the RING-type, the HECT-type, and the RING-between-RING (RBR) type [14,22,23]. RING-type E3 ubiquitin ligases, such as tumor necrosis factor receptor-associated factor 6 (TRAF6), directly catalyze ubiquitin transfer by recruiting E2 enzymes [14,24]. The HECT-type E3 ubiquitin ligases, such as WW domain-containing E3 ubiquitin-protein ligase 1 (WWP1), function by forming a ubiquitin–E3 intermediate [21]. The RBR-type E3 ubiquitin ligases have the characteristics of both the RING and HECT types [1]. The traditional K48-linked chains direct proteins to the proteasome for degradation, while K63 chains are involved in signal activation such as in the NF-κB pathway, and M1 chains regulate the assembly of inflammasomes [25].

In addition, the non-canonical ubiquitination pathway has received much attention. This term denotes modifications via non-lysine residues (N-terminal formylation and cysteine thiolation) or atypical lysine chain linkages (K27, M1) which differ from canonical K48/K63-linked degradation or signaling pathways [26,27]. It breaks with the traditional pattern. Unconventional types of post-translational modifications, such as N-terminal formylation, have emerged and play an important role in physiological and pathological processes, especially in inflammation [28,29]. During sepsis, non-canonical ubiquitination can rapidly respond to inflammation. For example, the HECT domain-containing ubiquitin E3 ligase HUWE1 (HUWE1) modifies NOD-like receptor family pyrin domain-containing protein 3 (NLRP3) through non-K27 chains to regulate inflammation [30]. In the inflammatory signaling pathway, the mechanisms of ubiquitination are diverse and complex. For instance, the TNF-induced inflammatory pathway is regulated by both canonical (K48 linkage) and non-canonical (M1, K11 linkage) codes [31]. The binding of TNF to the tumor necrosis factor receptor (TNFR) triggers the formation of the TNFR complex I. In the initial stage, cellular inhibitors of apoptosis proteins (cIAPs) and RIPK1 are ubiquitinated, and then related factors such as TNFR1/TRAF2 are recruited [31,32,33,34]. The catalysis of ubiquitin esterification by HOIL-1 is crucial [35]. HOIL-1L catalyzes the monoubiquitination of LUBAC subunits to enable HOIP-mediated linear ubiquitination, which attenuates LUBAC function and regulates NF-κB signaling [35]. Downstream, phosphorylation of the inhibitor of nuclear factor kappa-b alpha (IκB-α) at Ser32 and Ser36 leads to its K48-linked ubiquitination and degradation, promoting the nuclear translocation of NF-κB [36]. The ubiquitination of RIPK1 in different complexes can induce apoptosis or necroptosis [37]. In the IL-1-mediated pathway, the mixed chains of K11 and K63 regulate NF-κB, expanding the repertoire of ubiquitin signal transduction [38]. Non-canonical ubiquitination is also crucial in other physiological processes. For example, in Parkin-mediated mitophagy, damaged mitochondria can recruit the kinase PINK1 and the E3 ligase Parkin [39]. This process induces the PINK1-dependent phosphorylation of ubiquitin and Parkin [39]. Together with the regulation of ubiquitination in inflammation, this reflects the wide role of ubiquitination in cell physiology and pathology [31].

## 3. Ubiquitination in Sepsis and the Inflammatory Response

### 3.1. Regulation of the Production of Inflammatory Cytokines

The abnormal expression of inflammatory cytokines is one of the core pathological mechanisms of inflammatory diseases such as sepsis. The NF-κB signaling pathway, as a key hub for the transcription of inflammatory cytokines, has its activity precisely regulated by ubiquitination modifications [40]. Upon the stimulation of toll-like receptors (TLRs) or TNFRs, ubiquitination activates the NF-κB signaling pathway through the non-degradative effects of K63/M1-type polyubiquitin chains (Figure 2) [41]. After TLR activation, myeloid differentiation primary response 88 (MyD88) and IL-1 receptor-associated kinase 1/4 (IRAK1/4) are recruited to bind to TRAF6 and undergo K63-type polyubiquitination with the cooperation of the ubiquitin-conjugating enzyme Ubc13 [20]. This modification does not rely on protein degradation but serves as a signaling platform to recruit downstream kinases such as transforming growth factor-β-activated kinase 1 (TAK1), initiating the NF-κB activation program [20]. TNFR stimulation depends on LUBAC to extend K63-linked ubiquitination on the NF-κB essential modulator (NEMO) with linear M1-type ubiquitin chains [41,42,43,44]. This modification recruits the IκB kinase (IKK) complex through NEMO, leading to the release of the NF-κB transcription factors (p50/p65) from their cytoplasmic inhibitory complex with IκBα, which then drives the transcription of inflammatory cytokines such as TNF-α and interleukin-1β (IL-1β) [41,45,46,47]. It is worth noting that this K63/M1-type ubiquitination belongs to a non-degradative modification, and its core function is to promote the nuclear translocation of NF-κB by constructing a signaling complex, rather than inducing the degradation of target proteins [28]. In addition, deubiquitinating enzymes (DUBs) such as OTU DUB with linear linkage specificity (OTULIN), cylindromatosis (CYLD), and A20 limit the excessive activation of NF-κB by removing ubiquitin chains [41,46,47].

The traditional view is that ubiquitination inhibits NF-κB by mediating the proteasomal degradation of p65 through K48-type chains [48]. However, the latest research has found that in the lipopolysaccharide (LPS)-induced sepsis model, van Gogh-like protein 2 (VANGL2) recruits PDZ and lim domain protein 2 (PDLIM2) to catalyze the K63-type ubiquitination of p65 [49]. The nuclear dot protein 52 kDa (NDP52) transports p65 to the autolysosome for degradation by recognizing the K63 chain, and this process does not rely on the proteasome, reflecting the specific association between the type of ubiquitination modification and the degradation pathway [49].

### 3.2. Regulation of the Activation of Inflammasomes

Inflammasomes play a key role in the inflammatory response of sepsis and the NLRP3 inflammasome is one of the most widely studied types [50]. Recent studies have shown that the protein ubiquitination system is involved in regulating the activation process of the NLRP3 inflammasome, and this mechanism is of great significance for the development of sepsis [16,51]. Ubiquitination and deubiquitination modifications can play a protective role in inflammatory-related diseases by regulating various pathological processes such as excessive inflammatory responses, pyroptosis, abnormal autophagy, proliferation disorders, and oxidative stress injuries [21]. Taking the non-canonical ubiquitination regulation mechanism as an example, WWP1, an E3 ubiquitin ligase, has a dual role. Although its overexpression promotes NLRP3 ubiquitination, it simultaneously suppresses NLRP3 inflammasome activation and caspase-1-dependent gasdermin D (GSDMD) cleavage [52,53,54]. In addition, WWP1 is downregulated in sepsis [52]. It can promote the proteasomal degradation of TRAF6 by inducing K48-linked polyubiquitination, negatively regulating the release of TNF-α and IL-6 mediated by TLR4 [52,55].

HUWE1, an HECT-type E3 ligase, shows specific regulatory functions. In an LPS- and ATP-induced mouse bone marrow-derived macrophage (BMDM) model, it directly interacts with the NACHT domain of NLRP3, absent in melanoma 2 (AIM2), and NLRC4 via its BH3 domain [30]. This interaction triggers non-lysine-dependent K27-linked polyubiquitination—a non-canonical modification often misclassified as “non-lysine-dependent” [14]. Notably, K27 ubiquitination occurs at lysine residues but uses an atypical linkage, distinct from non-lysine modifications (N-terminal ubiquitination) [45]. This modification does not mediate protein degradation but promotes the assembly of the inflammasome by inducing a conformational change in NLRP3, thereby enhancing the maturation of caspase-1 and the release of downstream pro-inflammatory factors such as IL-1β [17].

Studies show that HUWE1 interacts with the NACHT domain of NLRP3 via its BH3 motif, mediating K27-linked polyubiquitination to promote inflammasome assembly, apoptosis-associated speck-like protein containing a CARD (ASC) speck formation, and caspase-1 activation [30,56]. HUWE1-deficient BMDMs exhibit reduced caspase-1 maturation and IL-1β secretion while inhibition with BI8622 suppresses NLRP3 activation in mouse and human cells [30]. Mechanistically, non-proteolytic K27 ubiquitination disrupts the autoinhibitory interaction of NLRP3 with leucine-rich repeat and pyrin domain-containing protein (LRR-PYD) via steric hindrance, driving a conformational transition to an oligomerizable state—corroborated by ASC speck formation [57]. The conserved role of HUWE1 in AIM2 and NLR family CARD domain-containing protein 4 (NLRC4) inflammasomes suggests that K27 ubiquitination is a unifying mechanism for inflammasome conformational activation [30].

Ubiquitin-specific peptidase 22 (USP22) negatively regulates NLRP3. It degrades NLRP3 through the autophagy-related 5 (ATG5)-mediated autophagy pathway, thereby suppressing inflammasome activation [58,59]. Mechanistically, USP22 stabilizes ATG5 by reducing the K27- and K48-linked ubiquitination of ATG5 at the K118 site [58]. In vivo studies have shown that the deficiency or silencing of USP22 significantly exacerbates peritonitis induced by alum and the systemic inflammation induced by LPS [58]. In summary, targeting the excessive activation of the NLRP3 inflammasome provides a potential strategy for the prevention or treatment of inflammatory-related diseases such as sepsis, and in-depth analysis of the ubiquitination modification mechanism will lay the foundation for the development of new therapeutic targets [57,60,61,62]. Table 1 summarizes the non-traditional roles of protein ubiquitination in sepsis pathogenesis, integrating the regulatory mechanisms of inflammatory responses, immune cell functions, and organ protection discussed above.

## 4. Ubiquitination in Sepsis and Immune Cell Functions

### 4.1. Activation of Neutrophils

Neutrophils, acting as the primary immunological defense in sepsis, experience complex functional control by the ubiquitination network [84]. Heparin-binding protein (HBP), released during sepsis, obstructs the K48-linked ubiquitination of the E3 ubiquitin ligase TRIM21, thereby stabilizing TRIM21 and facilitating the K63-linked ubiquitination of transcription factor p65 [63]. This process mediates pulmonary microvascular endothelial hyperpermeability and glycolytic dysfunction through the TRIM21-p65 signaling axis, thus contributing to the pathogenesis of acute lung injury (ALI) [63]. Mechanistically, heat shock protein Hsp90 modulates neutrophil apoptosis by maintaining the c-Src/caspase-8 complex, hence inhibiting its ubiquitination [85]. The E3 ligase Mid1 facilitates neutrophil–endothelial adhesion by downregulating the protein phosphatase 2Ac (PP2Ac), hence promoting the intercellular adhesion molecule-1 (ICAM-1) expression, whereas silencing Mid1 mitigates septic lung injury by inhibiting the Mid1-PP2Ac axis [86]. These data collectively indicate that ubiquitination alterations play a crucial role in sepsis-induced ALI by modulating neutrophil activation, apoptosis, and endothelial interactions, underscoring the ubiquitination network as a potential therapeutic target for sepsis-related lung injury.

### 4.2. Macrophage Polarization

In the pathogenesis of sepsis, the balance of M1/M2 macrophage polarization is a key type of immune regulation, and its state directly affects the outcome of the disease [87]. Macrophages polarize into a pro-inflammatory M1 type and an anti-inflammatory M2 type [88,89]. The dynamic imbalance between the two is closely related to the progression of sepsis, but its molecular mechanism has not been fully elucidated [90,91]. Studies have shown that the level of malignant fibrous histiocytoma amplified sequence 1 (MFHAS1) in sepsis patients is significantly increased, and it drives the inflammatory response by activating the TLR2/c-Jun N-terminal kinase (JNK)/NF-κB pathway (Figure 3) [64]. The E3 ubiquitin ligase praja ring finger 2 (Praja2) can bind to MFHAS1 and mediate its non-degradative ubiquitination, promoting MFHAS1 accumulation [64]. This process enhances the activation of the TLR2-mediated JNK/p38 signaling pathway, driving macrophage polarization from the M2 to the M1 phenotype and exacerbating the inflammatory response [64]. Praja2-ubiquitinated MFHAS1 drives macrophage M1 polarization via the TLR2-JNK/p38 axis, upregulating M1 markers (IL-6, TNF-α, IL-1β, and inducible nitric oxide synthase (iNOS)) and downregulating M2 markers (IL-6, iNOS, 1L-10, Arginase-1, and macrophage mannose receptor (MMR)) [65,66]. The blockade of JNK specifically inhibits M1 marker expression, whereas p38 inhibition exclusively affects M1 and M2 markers. This ubiquitination activates NF-κB to promote JNK/p38 signaling [89,92], and Pam3CSK4 stimulation enhances Ly6C^+^ M1 polarization, as confirmed by functional assays [64]. Collectively, these findings indicate that this mechanism exacerbates inflammation [91]. Other studies have further revealed the diversity of ubiquitination regulation, exemplified by the interaction between A20 and the NIMA-related kinase 7 (NEK7) [67]. A20 directly binds to NEK7, promotes its proteasomal degradation via enhanced ubiquitination with key functional sites at K189 and K293 residues of NEK7, and inhibits NEK7 binding to the NLRP3 complex through its OTU domain and ZnF4/ZnF7 motifs [67,68]. Interfering with the function of NEK7 in macrophages can significantly inhibit pyroptosis and alleviate the process of sepsis [67].

The regulatory role of ubiquitin-conjugating enzyme E2 M (UBE2M) is evident in an *Escherichia coli*-induced sepsis mouse model. Specifically, the macrophage-specific deletion of UBE2M—a key enzyme for ubiquitination modification—reduces levels of pro-inflammatory cytokines (IL-1β, IL-6, and TNF-α) and organ damage, improves the survival rate, and does not affect bacterial clearance [69]. Mechanistically, the deletion of UBE2M inhibits the activation of the NF-κB, ERK, and JAK-STAT signaling pathways, downregulating the excessive inflammatory response [69].

### 4.3. Regulation of T Cell Functions

T cells play an important role in the immune response of sepsis and ubiquitination affects their functions by regulating the activation, proliferation, differentiation, and apoptosis of T cells [93,94,95]. Among them, casitas B lymphoma-b (Cbl-b) is a key downstream regulator of the CD28 co-stimulation and cytotoxic-lymphocyte-associated protein 4 (CTLA-4) co-inhibition signaling pathways, and E3 ubiquitin ligases play a central role in the regulation of effector T cell functions [70]. Cbl-b, through multiple protein interaction domains such as binding to TCR signaling molecules such as lymphocyte-specific tyrosine kinase (LCK), SH2 domain-containing leukocyte protein of 76kD (SLP76), and zeta-associated protein of 70 kDa (ZAP70), cooperates with the E3 ligase Itch to mediate the polyubiquitination of the K33 site of the TCR-ζ subunit [71]. This modification does not induce the degradation or endocytosis of TCR but inhibits the activation of T cells by preventing the phosphorylation of TCR and its binding to the downstream ZAP70 kinase. This process does not require CD28 co-stimulation [71,72]. It is worth noting that Cbl-b knockout mice exhibit the excessive activation of T cells, independent of CD28 stimulation, but no autoimmune damage has been observed [73,74]. This characteristic makes Cbl-b a potential target for cancer immunotherapy as it has multiple immune checkpoint inhibitory functions and a relatively low risk of autoimmune toxicity [72].

## 5. Ubiquitination in Sepsis and Organ Protection

### 5.1. Lung Protection

The lungs are one of the organs most vulnerable to sepsis, and their injury and fibrosis processes are closely related to ubiquitination regulation [96]. Studies have shown that in LPS-induced sepsis mice, the E3 ubiquitin ligase TRIM27 is significantly upregulated and positively correlated with the degree of lung injury [19]. Knocking down TRIM27 can inhibit the ubiquitination degradation of PPARγ; reduce the expression of NADPH oxidase 4 (NOX4) and the activation of the downstream p-p65 inflammatory pathway; and alleviate the inflammatory infiltration, apoptosis, and oxidative stress injury of lung tissues [19]. Overexpressing NOX4 can reverse this protective effect, revealing the key role of the “TRIM27-PPARγ-NOX4” ubiquitination axis in the lung injury of sepsis [19]. In addition, pulmonary fibrosis in sepsis is related to ubiquitination regulation [96]. The E3 ubiquitin ligase TRAF6 and the deubiquitinating enzyme USP38 regulate interleukin 33 receptor (IL-33R) levels and signal transduction via K27-linked polyubiquitination and deubiquitination [75].

### 5.2. Liver Protection

Liver injury induced by sepsis is a common complication. Although the liver has a relatively low incidence of failure due to its strong regenerative and anti-inflammatory capabilities, the mortality rate of septic patients with liver failure is high [97]. Therefore, there is an urgent need to develop theories and treatment methods for liver protection and the prevention of liver failure. Studies have shown that ubiquitination is involved in regulating the process of liver injury in sepsis [98]. It exerts its effects by regulating hepatocyte metabolism, functions, and responses to oxidative stress [99,100]. Some ubiquitin-related proteins can protect hepatocytes from the damage of inflammatory cytokines by degrading damaged proteins or activating antioxidant signaling pathways. For example, OTUD1 reduces oxidative stress, apoptosis, and inflammation induced by liver ischemia/reperfusion (I/R) injury [76]. Mechanistically, OTUD1 deubiquitinates and activates nuclear factor erythroid 2-related factor 2 (NRF2) through its catalytic cysteine 320 residues and the ETGE motif, thereby alleviating liver I/R injury [76].

In the LPS-induced sepsis model, the excessive activation of poly (ADP-ribose) polymerase 1 (PARP1) in macrophages is a key factor driving the inflammatory response [101]. Pimpinellin can upregulate the E3 ubiquitin ligase ring finger protein 146 (RNF146), promote K48-linked ubiquitination modification, and target PARP1 for degradation [77,78]. This inhibits the release of pro-inflammatory factors such as TNF-α and IL-6 by macrophages and significantly reduces the inflammatory infiltration, apoptosis, and oxidative stress injury of hepatocytes induced by LPS [77]. This protective effect is mediated by the PARP1 ubiquitination degradation pathway [77]. Knocking out PARP1 abolishes the ameliorative effect of specific intervention on liver injury in sepsis [77]. Mechanistically, pimpinellin enhances the ubiquitination-mediated degradation of PARP1, blocks the parthanatos cell death pathway, and restores mitochondrial function, providing a new intervention target of the “RNF146/PARP1 ubiquitination axis” for the treatment of liver injury in sepsis [77]. In addition, a decrease in the level of deubiquitinase USP4 will exacerbate liver inflammation and fibrosis [102,103].

### 5.3. Cardiac Function

SIMD is a severe complication of sepsis, characterized by impaired cardiac function and a high mortality rate [10]. Ubiquitination and deubiquitination, as key PTMs, are involved in crucial cellular processes such as inflammation, apoptosis, mitochondrial function, and calcium handling by regulating protein stability, localization, and activity [1,104,105,106]. The dysregulation of the ubiquitination and deubiquitination systems has been gradually confirmed to be closely related to the pathogenesis of SIMD [1]. Dysfunction of the ubiquitin–proteasome system (UPS) is often driven by changes in the activity of E3 ligases, which accelerate the degradation of key regulatory proteins and exacerbate cardiac inflammation, oxidative stress, and apoptosis [1,79]. The imbalance of DUB activity disrupts protein homeostasis and further amplifies myocardial damage [1]. For example, USP7 can stabilize the transcription factor SOX9 through deubiquitination and upregulate its protein expression [79]. SOX9 inhibits the expression of miR-96-5p by binding to its promoter region and then promotes the expression of NLRP3, exacerbating the myocardial injury and pyroptosis of cardiomyocytes induced by sepsis [79].

### 5.4. Renal Function

More than half of critically ill septic patients will develop acute kidney injury (AKI), which significantly increases the risk of death [107]. The pro-inflammatory effect of macrophages can exacerbate tubular injury in the early stage of AKI [108]. The ubiquitination mechanism is involved in the protection of AKI by regulating macrophage functions and mitophagy [109,110]. E3 ubiquitin ligases MARCHF1 and MARCHF8 can ubiquitinate the T cell activation molecule 1 (TARM1) on the surface of myeloid cells, induce its internalization, and degrade it in phagolysosomes, thereby inhibiting excessive renal inflammation and reducing AKI [110]. Meanwhile, the PINK1/PARK2 pathway in renal cells recruits and phosphorylates to activate the E3 ligase activity of PARK2 through PINK1, prompting the ubiquitination labeling of damaged mitochondria and their phagocytosis and degradation by autophagosomes, alleviating sepsis-related AKI through mitophagy [80].

### 5.5. Intestinal Function

The intestine is an important target organ for immune regulation in sepsis, and its dysfunction is closely related to the gut microbiota, immune cells, and the ubiquitination mechanism [111]. A large number of immune cells and microorganisms inhabit the intestine, and the metabolites derived from the microbiota can maintain the immune homeostasis of the intestine and the whole body [112]. Sepsis can induce an increase in apoptosis, a decrease in proliferation, and a decline in the migration ability of intestinal epithelial cells, thereby disrupting the intestinal mucosal barrier [113,114]. Studies have shown that USP47 is involved in the occurrence of intestinal injury in sepsis by regulating the inflammatory signaling pathway in intestinal epithelial cells [81,82]. USP47 removes the K63-type polyubiquitination modification of TRAF6, stabilizes the TRAF6 protein, and activates its downstream inflammatory pathway, thus exacerbating the intestinal inflammatory response [81,83]. This mechanism reveals the crucial role of ubiquitination modification in the intestinal immune dysregulation of sepsis and provides a potential intervention direction for targeting intestinal inflammation.

## 6. Conclusions

Protein ubiquitination plays multiple non-traditional roles in sepsis, involving the regulation of the inflammatory response, modulation of immune cell functions, and protection of organs. These non-traditional roles are different from the classical functions of ubiquitination in protein degradation and cell cycle control, providing a new perspective for understanding the pathogenesis of sepsis. Exploring the molecular mechanisms of protein ubiquitination in sepsis is conducive to the discovery of new therapeutic targets and is of great significance for the development of new treatment strategies for sepsis. In the future, more in-depth research is needed to elucidate the complex network of ubiquitination in sepsis and provide more effective assistance for clinical treatment.

## Figures and Tables

**Figure 1 cells-14-01012-f001:**
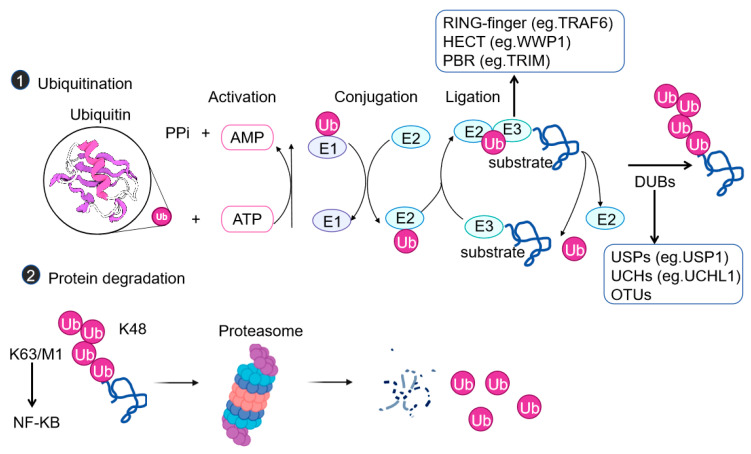
Mechanisms and functions of the ubiquitin system. The ubiquitination cascade initiates with the ATP-dependent activation of ubiquitin by an E1-activating enzyme. The activated ubiquitin is then shuttled to an E2-conjugating enzyme, which serves as an intermediate carrier. E3 ligases play a crucial role in catalyzing the transfer of ubiquitin from E2 to specific substrate proteins. Conversely, deubiquitinating enzymes (DUBs) counteract this post-translational modification by hydrolyzing and removing ubiquitin moieties from modified substrates. Enzymatic steps and enzymes are involved in protein ubiquitination, a reversible and versatile PTM. Lysine 48 (K48)- and lysine 63 (K63)-linked polyubiquitin chains are abundant and well-studied. For other types of chains, many are referred to as atypical chains. AMP, adenosine monophosphate; ATP, adenosine triphosphate; DUB, deubiquitinase; E1, ubiquitin-activating enzyme; E2, ubiquitin-conjugating enzyme; E3, ubiquitin ligase; K, lysine; M, methionine; OTUs, ovarian tumor deubiquitinases; PPi, inorganic pyrophosphate; TGF-β, transforming growth factor-β; Ub, ubiquitin; USPs, ubiquitin-specific proteases; UCHs, ubiquitin C-terminal hydrolases. Illustration created using BioRender (biorender.com) uses R version 4.2.2.

**Figure 2 cells-14-01012-f002:**
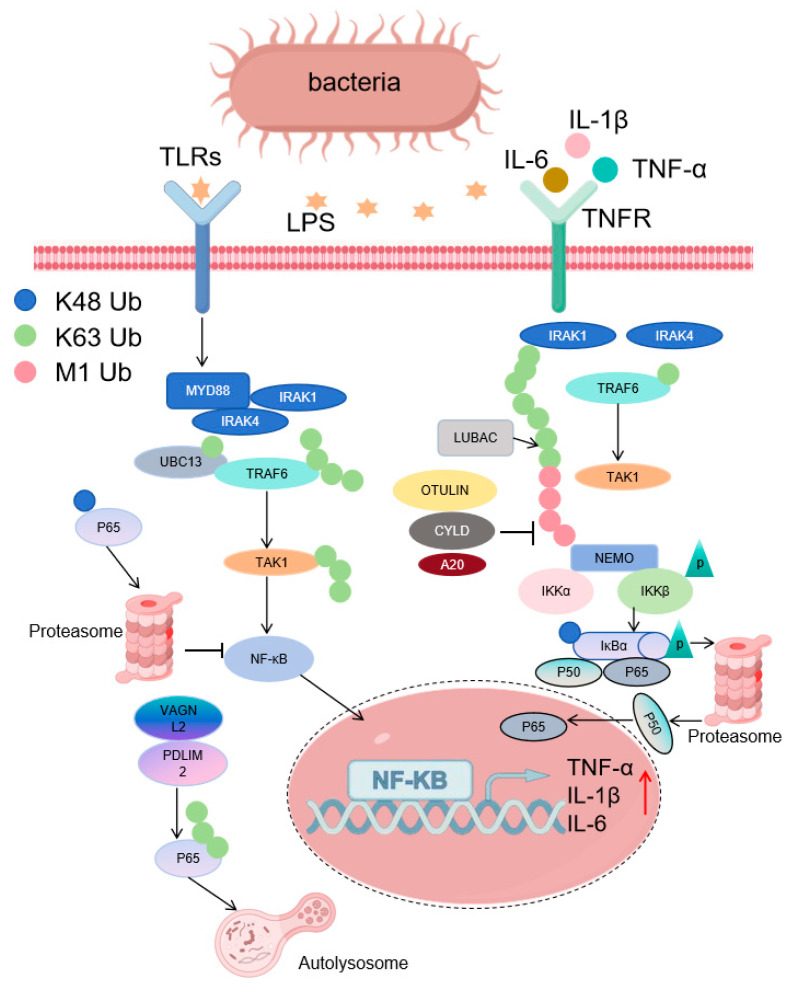
The impact of ubiquitination on inflammatory signaling pathways in sepsis. Ubiquitination regulates TLRs and TNFRs and their downstream inflammatory pathways during sepsis. Upon receptor stimulation, the K63-linked polyubiquitination of MyD88/IRAK1/4 (via Ubc13) or M1-linked polyubiquitination of NEMO (via LUBAC) recruits TAK1 or the IKK complex, respectively, driving NF-κB nuclear translocation and transcription of cytokines (TNF-α, IL-1β). Additionally, VANGL2 recruits PDLIM2 to catalyze the K63-type ubiquitination of p65 in LPS-induced sepsis, and NDP52 recognizes K63 chains to mediate p65 autolysosomal degradation, independent of the proteasome. Provided by FigDraw.

**Figure 3 cells-14-01012-f003:**
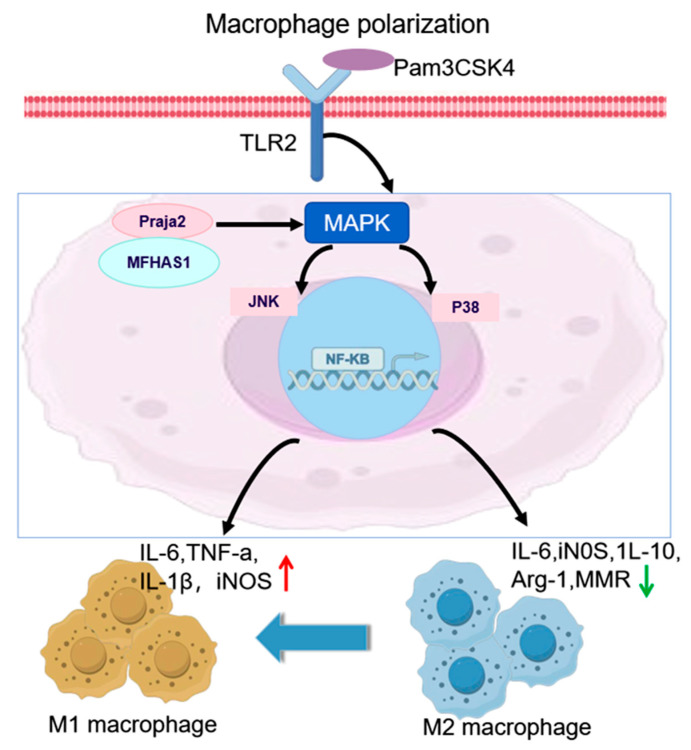
Mechanistic model of MFHAS1 ubiquitylation-mediated regulation of the MAPK pathway during macrophage polarization. Ubiquitylation of MFHAS1 by Praja2 activates the TLR2/JNK/p38/NF-κB signaling cascade, driving M1 macrophage polarization and promoting M2-to-M1 phenotypic transition [64]. Provided by FigDraw.

**Table 1 cells-14-01012-t001:** Non-traditional roles of protein ubiquitination in sepsis pathogenesis.

Pathological Process	Key Ubiquitination Event	Enzyme/Complex Involved	Mechanism	Functional Impact in Sepsis	Refs.
Inflammatory response	NF-κB non-degradative activation	TRAF6, LUBAC, Ubc13	K63/M1 polyubiquitination of TRAF6/RIPK1 recruits the TAK1/IKK complex, promoting NF-κB nuclear translocation without protein degradation	Enhances TNF-α/IL-1β production, driving systemic inflammation	[41,42,43,44,45,46,47]
	NLRP3 inflammasome assembly	HUWE1, WWP1, USP22	HUWE1-mediated K27 ubiquitination induces NLRP3 conformational change for inflammasome assembly; WWP1/USP22 regulates via degradation or autophagy	Promotes IL-1β release (HUWE1) or inhibits pyroptosis (USP22)	[30,52,53,54,55,56,57]
Immune cell functions	Activation of neutrophils	TRIM21, HBP	Inhibition of K48-linked ubiquitination of TRIM21; promotion of K63-linked ubiquitination of p65	Contributes to acute lung injury (ALI) pathogenesis in sepsis	[63]
	Macrophage M1/M2 polarization	Praja2, A20, UBE2M	Praja2 enhances MFHAS1 accumulation via non-degradative ubiquitination, driving M1 polarization; A20 degrades NEK7 to inhibit NLRP3	M1-dominated inflammation (Praja2) or reduced pyroptosis (A20)	[64,65,66,67,68,69]
	T cell activation inhibition	Cbl-b, Itch	Cbl-b/Itch mediates K33 ubiquitination of TCR-ζ, blocking ZAP70 recruitment and T cell activation	Suppressed excessive T cell response, preventing immunopathology	[70,71,72,73,74]
Organ protection	Lung injury regulation	TRIM27, TRAF6, USP38	TRIM27 promotes PPARγ degradation via K48 ubiquitination, exacerbating NOX4-mediated oxidative stress; TRAF6/USP38 regulates IL-33R signaling	Enhanced oxidative stress (TRIM27) or fibrosis (TRAF6/USP38)	[19,75]
	Liver anti-oxidative stress	OTUD1, RNF146	OTUD1 deubiquitinates NRF2 to activate antioxidant pathways; RNF146 promotes PARP1 degradation via K48 ubiquitination	Reduced hepatic oxidative injury and parthanatos (RNF146)	[76,77,78]
	Cardiomyocyte pyroptosis	USP7, SOX9	USP7 stabilizes SOX9 via deubiquitination, upregulating miR-96-5p and NLRP3 expression	Exacerbated myocardial pyroptosis and dysfunction	[79]
	Renal mitophagy activation	PINK1/PARK2 pathway	PARK2-mediated ubiquitination of damaged mitochondria promotes mitophagy and reduces tubular injury	Alleviated acute kidney injury (AKI) via mitochondrial clearance	[80]
	Intestinal barrier protection	USP47	USP47 stabilizes TRAF6 via deubiquitination, enhancing NF-κB-driven intestinal inflammation	Disrupted mucosal barrier and enhanced gut inflammation	[81,82,83]

## Data Availability

Not applicable.

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
