# Peer review of "The Role of Protein Ubiquitination in the Onset and Progression of Sepsis"

_cells, 2025, doi:10.3390/cells14131012_

Round 1

Reviewer 1 Report

Comments and Suggestions for Authors

Major points:

1) The article lacks sufficient number of visual outputs namely represented by figures and tables. Please provide at least three more figures/tables in addition to Figure 1 to better illustrate the developing story and highlight its key concepts in an illustrative and easy to digest manner.

2) There seem to be many references missing leaving the vast proportion of text uncited. Please see the minor points below for concrete examples.

Minor points:

1) Please change "disease" to something like "condition" (lines 9, 25).

2) Please replace "dysfunction[1]" with "dysfunction [1]" (line 26).

3) Please change "year[2, 3]" to "year [2, 3]" (line 29).

4) Please replace "20%-30%" with "20–30%" (line 29).

5) Please change "(MODS)[4]" to "(MODS) [4]" (line 33).

6) Please replace "kidneys[5, 6]" with "kidneys [5, 6]" (line 35).

7) Please change "critical[7, 8]" to "critical [7, 8]" (line 36).

8) Please replace "13.8%-40%" with "13.8–40%" (line 37).

9) Please change "70%-90%" to "70–90%" (line 38).

10) Please replace "health[9, 10]" with "health [9, 10]" (line 38).

11) Please provide reference for "Although its pathological mechanism has not been fully elucidated, recent studies have shown that protein ubiquitination, a non-traditional post-translational modification, plays a central regulatory role in the immune imbalance and organ injury in sepsis" (line 38).

12) Please provide reference for "Protein ubiquitination, through the E1-E2-E3 enzyme cascade reaction, attaches ubiquitin molecules (76 amino acids) to target proteins via isopeptide bonds (lysine residues) or unconventional bonds (N-terminus, cysteine, etc.), forming various chain types such as K48 (for proteasomal degradation), K63 (for signal activation), and M1 (linear chain, for inflammatory regulation)" (line 42).

13) Please provide reference for "Different from the traditionally recognized "proteasomal degradation function", recent studies have revealed that in sepsis, it regulates the inflammatory signaling pathway, immune cell polarization, and organ protection mechanisms through non-degradative modifications (such as K63/M1 chains)" (line 46).

14) Please change "cardiomyocytes[11, 12]" to "cardiomyocytes [11, 12]" (line 53).

15) Please replace "ubiquitination[13]" with "ubiquitination [13]" (line 54).

16) Please provide reference for "Ubiquitination is a post-translational modification in which one or usually a chain of small proteins, ubiquitin chains composed of 76 amino acids, are covalently linked to the lysine residues within the substrate proteins" (line 58).

17) Please change "proteins[14]" to "proteins [14]" (line 62).

18) Please replace "ligases:" with "ligases," (line 65).

19) Please change "inflammasomes[15]" to "inflammasomes [15]" (line 71).

20) "fig1" sign appears when hovering the mouse cursor over Figure 1. Please disable this feature.

21) Please replace "Functions of the Ubiquitin System" with "functions of the ubiquitin system" (line 73).

22) Please change "well-studied; for" to "well-studied. For" (line 75).

23) Please replace "Ub: Ubiquitin; E1: Ubiquitin-activating enzyme; E2: Ubiquitin-conjugating enzyme; E3: Ubiquitin ligase; DUB: Deubiquitinase; AMP: Adenosine monophosphate; ATP: Adenosine triphosphate; PPi: Inorganic pyrophosphate; K: Lysine; M: Methionine; TGF-β: Transforming growth factor-β" with "AMP, adenosine monophosphate; ATP, adenosine triphosphate; DUBs, deubiquitinases; E1, ubiquitin-activating enzyme; E2, ubiquitin-conjugating enzyme; E3, ubiquitin ligase; K, lysine; M, methionine; PPi, inorganic pyrophosphate; TGF-β, transforming growth factor-β; Ub, ubiquitin" (line 76).

24) Please provide reference for "It breaks through the traditional pattern, and unconventional types such as N-terminal formylation have emerged, playing an important role in physiological and pathological processes, especially in inflammation" (line 82).

25) Please change "inflammation[16]" to "inflammation [16]" (line 86).

26) Please replace "codes[17]" with "codes [17]" (line 89).

27) Please change "TNFR" to "tumor necrosis factor receptors (TNFRs)" (line 90) and "tumor necrosis factor receptors (TNFRs)" to "TNFRs" (line 108).

28) Please change "crucial[18]" to "crucial [18]" (line 92).

29) Please provide reference for "Downstream, the phosphorylation of IκB-α leads to its K48-linked ubiquitination and degradation, promoting the nuclear translocation of NF-κB" (line 92).

30) Please replace "The ubiquitination" with "Ubiquitination" (line 94).

31) Please change "necroptosis[19]" to "necroptosis [19]" (line 95).

32) Please replace "transduction[20]" with "transduction [20]" (line 96).

33) "For example, in Parkin-mediated mitophagy, damaged mitochondria can recruit the kinase PINK1 and the E3 ligase PARKIN, inducing the PINK1-dependent phosphorylation of ubiquitin and PARKIN[21], which, together with the regulation of ubiquitination in inflammation, reflects its wide role in the physiological and pathological processes of cells" (line 97) is way too long. Please split into at least two sentences.

34) Please provide reference for "For example, in Parkin-mediated mitophagy, damaged mitochondria can recruit the kinase PINK1 and the E3 ligase PARKIN, inducing the PINK1-dependent phosphorylation of ubiquitin and PARKIN[21], which, together with the regulation of ubiquitination in inflammation, reflects its wide role in the physiological and pathological processes of cells" (line 97).

35) Please change "PARKIN" to "Parkin" (line 99).

36) Please replace "PARKIN[21]" with "Parkin [21]" (line 99).

37) Please change "nuclear factor κB (NF-κB)" to "NF-κB" (line 105).

38) Please provide reference for "Upon stimulation of Toll-like receptors (TLRs) or tumor necrosis factor receptors (TNFRs), ubiquitination activates the NF-κB signaling pathway through the non-degradative effects of K63/M1-type polyubiquitin chains" (line 107).

39) Please provide reference for "After TLR activation, MyD88 and IRAK1/4 are recruited, which then bind to TRAF6 and undergo K63-type polyubiquitination with the cooperation of Ubc13" (line 110).

40) Please provide reference for "This modification does not rely on protein degradation but serves as a signaling platform to recruit downstream kinases (such as TAK1), initiating the NF-κB activation program" (line 112).

41) Please replace "IL-1β[22-25]" with "IL-1β [22-25]" (line 117).

42) Please change "proteins[26]" to "proteins [26]" (line 120).

43) Please replace "OTULIN (an OUT deubiquitinase with linear bond specificity), CYLD (cylindromatosis)" with "OUT deubiquitinase with linear bond specificity (OTULIN), cylindromatosis (CYLD)" (line 121).

44) Please change "chains[23-25]" to "chains [23-25]" (line 123).

45) Please provide reference for "The traditional view is that ubiquitination inhibits NF-κB by mediating the proteasomal degradation of p65 through K48-type chains" (line 124).

46) Please provide reference for "However, the latest research has found that in the LPS-induced sepsis model, VANGL2 recruits PDLIM2 to catalyze the K63-type ubiquitination of p65" (line 125).

47) Please define abbreviation for "LPS" in "LPS-induced" (line 126).

48) Please replace "pathway[27]" with "pathway [27]" (line 130).

49) Please provide reference for "Recent studies have shown that the protein ubiquitination system is involved in regulating the activation process of the NLRP3 inflammasome, and this mechanism is of great significance for the development of sepsis" (line 134).

50) Please change "injuries[14]" to "injuries [14]" (line 140).

51) Please replace "example:" with "example." (line 141).

52) "The dual role of WWP1:" does not seem to fit the sentence "The dual role of WWP1: Although the overexpression of WWP1, an E3 ubiquitin ligase, can promote the ubiquitination of NLRP3, it can simultaneously inhibit the activation of the NLRP3 inflammasome and the cleavage of GSDMD mediated by caspase-1" (line 142). Please rephrase.

53) Please provide reference for "The dual role of WWP1: Although the overexpression of WWP1, an E3 ubiquitin ligase, can promote the ubiquitination of NLRP3, it can simultaneously inhibit the activation of the NLRP3 inflammasome and the cleavage of GSDMD mediated by caspase-1" (line 142).

54) Please provide reference for "In addition, WWP1 is downregulated in sepsis" (line 145).

55) Please change "TLR4[28, 29]" to "TLR4 [28, 29]" (line 147).

56) "The specific regulation of HUWE1:" does not seem to fit the sentence "The specific regulation of HUWE1: In the LPS and ATP-induced mouse bone marrow-derived macrophage (BMDM) model, the HECT-type E3 ligase HUWE1 directly interacts with the NACHT domain of NLRP3 through its BH3 domain, triggering non-lysine-dependent K27-linked polyubiquitination modification" (line 148). Please correct.

57) Please provide reference for ""The specific regulation of HUWE1: In the LPS and ATP-induced mouse bone marrow-derived macrophage (BMDM) model, the HECT-type E3 ligase HUWE1 directly interacts with the NACHT domain of NLRP3 through its BH3 domain, triggering non-lysine-dependent K27-linked polyubiquitination modification" (line 148).

58) Please replace "Caspase-1" with "caspase-1" (line 154).

59) Please change "IL-1β[12]" to "IL-1β [12]" (line 154).

60) "The negative regulation of USP22:" does not seem to fit the sentence "The negative regulation of USP22: Ubiquitin-specific peptidase 22 (USP22) can degrade NLRP3 through the ATG5-mediated autophagy pathway, thereby inhibiting the activation of the inflammasome" (line 155). Please revise.

61) Please provide reference for "The negative regulation of USP22: Ubiquitin-specific peptidase 22 (USP22) can degrade NLRP3 through the ATG5-mediated autophagy pathway, thereby inhibiting the activation of the inflammasome" (line 155).

62) Please provide reference for "Mechanistically, USP22 stabilizes ATG5 by reducing the K27 and K48-linked ubiquitination of ATG5 at the Lys118 site" (line 157).

63) Please replace "lipopolysaccharide[30]" with "LPS [30]" (line 160).

64) Please provide reference for "The dynamic imbalance between the two is closely related to the progression of sepsis, but its molecular mechanism has not been fully elucidated" (line 169).

65) Please change "pathway[31]" to "pathway [31]" (line 174).

66) "The E3 ubiquitin ligase Praja2 can bind to MFHAS1 and mediate its non-degradative ubiquitination, promoting the accumulation of MFHAS1, and then enhancing the activation of the JNK/p38 signaling pathway mediated by TLR2, driving the polarization of macrophages from the M2 type to the M1 type and exacerbating the inflammatory response[31]" (line 174) is way too long. Please split into at least two sentences.

67) Please replace "response[31]" with "response [31]" (line 178).

68) "Other studies have further revealed the diversity of ubiquitination regulation: the interaction between A20 and NEK7: A20 can directly bind to NEK7, promote its proteasomal degradation through enhanced ubiquitination (the key functional sites are the K189 and K293 residues of NEK7), and inhibit the binding of NEK7 to the NLRP3 complex through the OTU domain and the ZnF 4/ZnF 7 motifs" (line 178) is way too long. Please split into at least two sentences.

69) "the interaction between A20 and NEK7:" does not seem to fit "Other studies have further revealed the diversity of ubiquitination regulation: the interaction between A20 and NEK7: A20 can directly bind to NEK7, promote its proteasomal degradation through enhanced ubiquitination (the key functional sites are the K189 and K293 residues of NEK7), and inhibit the binding of NEK7 to the NLRP3 complex through the OTU domain and the ZnF 4/ZnF 7 motifs" (line 178). Please fix.

70) Please change "regulation:" to "regulation," (line 179).

71) Please replace "sepsis[14, 32]" with "sepsis [14, 32]" (line 184).

72) "The regulatory role of UBE2M: In the Escherichia coli-induced sepsis mouse model, the specific deletion of UBE2M, a key enzyme for ubiquitination modification in macrophages, can reduce the levels of pro-inflammatory cytokines such as IL-1β, IL-6, and TNF-α, organ damage and improve the survival rate, without affecting the ability to clear bacteria" (line 185) is way too long. Please split into at least two sentences.

73) "The regulatory role of UBE2M:" does not seem to fit "The regulatory role of UBE2M: In the Escherichia coli-induced sepsis mouse model, the specific deletion of UBE2M, a key enzyme for ubiquitination modification in macrophages, can reduce the levels of pro-inflammatory cytokines such as IL-1β, IL-6, and TNF-α, organ damage and improve the survival rate, without affecting the ability to clear bacteria" (line 185). Please rephrase.

74) Please change "response[33]" to "response [33]" (line 191).

75) Please provide reference for "cells play an important role in the immune response of sepsis, and ubiquitination affects their functions by regulating the activation, proliferation, differentiation, and apoptosis of T cells" (line 193).

76) Please replace "Casitas" with "casitas" (line 195).

77) Please change "functions[34]" to "functions [34]" (line 197).

78) Please provide reference for "Cbl-b, through multiple protein interaction domains (such as binding to TCR signaling molecules such as LCK, SLP76, and ZAP70), cooperates with the E3 ligase Itch to mediate the polyubiquitination of the Lys33 site of the TCR-ζ subunit" (line 198).

79) Please replace "co-stimulation[35, 36]" with "co-stimulation [35, 36]" (line 203).

80) Please change "observed[37, 38]" to "observed [37, 38]" (line 205).

81) Please change "immunotherapy -" to "immunotherapy," (line 206).

82) Please provide reference for "The lung is one of the organs most vulnerable to sepsis, and its injury and fibrosis processes are closely related to ubiquitination regulation" (line 210).

83) Please provide reference for "Studies have shown that in LPS-induced sepsis mice, the E3 ubiquitin ligase TRIM27 is significantly upregulated and positively correlated with the degree of lung injury" (line 211).

84) Please provide reference for "Knocking down TRIM27 can inhibit the ubiquitination degradation of peroxisome proliferator-activated receptor γ (PPARγ), reduce the expression of NADPH oxidase 4 (NOX4) and the activation of the downstream p-p65 inflammatory pathway, alleviate the inflammatory infiltration, apoptosis, and oxidative stress injury of lung tissues." (line 213).

85) Please replace "effect[13]" with "effect [13]" (line 218).

86) Please provide reference for "In addition, pulmonary fibrosis in sepsis is related to ubiquitination regulation" (line 219).

87) "The E3 ubiquitin ligase tumor necrosis factor receptor-associated factor 6 (TRAF6) and the deubiquitinating enzyme USP38 regulate the level and signal transduction of the interleukin 33 receptor (IL-33R) through K27-linked polyubiquitination and deubiquitination, thereby affecting the inflammatory response and fibrosis of the lungs, providing a precise direction for related treatments[39]" (line 220) is way too long. Please split into at least two sentences.

88) Please change "treatments[39]" to "treatments [39]" (line 224).

89) Please replace "high[40]" with "high [40]" (line 228).

90) Please provide reference for "Studies have shown that ubiquitination is involved in regulating the process of liver injury in sepsis" (line 230).

91) Please provide reference for "It exerts its effects by regulating hepatocyte metabolism, functions, and responses to oxidative stress" (line 231).

92) Please provide reference for "For example, OTUD1 reduces oxidative stress, apoptosis, and inflammation induced by liver ischemia/reperfusion (I/R) injury" (line 234).

93) Please change "residues" to "residue" (line 237).

94) Please replace "injury[41]" with "injury [41]" (line 238).

95) Please provide reference for "In the LPS-induced sepsis model, the excessive activation of poly (ADP-ribose) polymerase 1 (PARP1) in macrophages is a key factor driving the inflammatory response" (line 239).

96) Please provide reference for "Pimpinellin can upregulate the E3 ubiquitin ligase RNF146, promote K48-linked ubiquitination modification, and target PARP1 for degradation" (line 241).

97) Please provide reference for "This inhibits the release of proinflammatory factors such as TNF-α and IL-6 by macrophages and significantly reduces the inflammatory infiltration, apoptosis, and oxidative stress injury of hepatocytes induced by LPS" (line 242).

98) It is not exactly clear what the authors mean by "impaneling on liver injury" in "This protective effect depends on the PARP1 ubiquitination degradation pathway, and knocking out PARP1 will eliminate the ameliorative effect of impaneling on liver injury in sepsis" (line 245)?

99) Please provide reference for "This protective effect depends on the PARP1 ubiquitination degradation pathway, and knocking out PARP1 will eliminate the ameliorative effect of impaneling on liver injury in sepsis" (line 245).

100) Please change "fibrosis[43, 44]" to "fibrosis [43, 44]" (line 251).

101) Please provide reference for "Ubiquitination and deubiquitination, as key post-translational modifications (PTMs), are involved in crucial cellular processes such as inflammation, apoptosis, mitochondrial function, and calcium handling by regulating protein stability, localization, and activity" (line 254).

102) Please provide reference for "The dysregulation of the ubiquitination and deubiquitination systems has been gradually confirmed to be closely related to the pathogenesis of SIMD" (line 257).

103) Please replace "The dysfunction" with "Dysfunction" (line 259).

104) Please provide reference for "The dysfunction of the ubiquitin-proteasome system (UPS) is often driven by changes in the activity of E3 ligases, which accelerate the degradation of key regulatory proteins and exacerbate cardiac inflammation, oxidative stress, and apoptosis" (line 259).

105) Please change "damage[1]" to "damage [1]" (line 263).

106) Please provide reference for "For example, ubiquitin-specific peptidase 7 (USP7) can stabilize the transcription factor SOX9 through deubiquitination and upregulate its protein expression" (line 263).

107) Please replace "sepsis[45]" with "sepsis [45]" (line 268).

108) Please change "death[46]" to "death [46]" (line 271).

109) Please provide reference for "The pro-inflammatory effect of macrophages can exacerbate tubular injury in the early stage of AKI" (line 271).

110) Please provide reference for "The ubiquitination mechanism is involved in the protection of AKI by regulating macrophage functions and mitophagy" (line 272).

111) Please replace "AKI[47]" with "AKI [47]" (line 277).

112) Please change "mitophagy[48]" to "mitophagy [48]" (line 281).

113) Please replace "body[49]" with "body [49]" (line 287).

114) Please provide reference for "Sepsis can induce an increase in apoptosis, a decrease in proliferation, and a decline in the migration ability of intestinal epithelial cells, thereby disrupting the intestinal mucosal barrier" (line 287).

115) Please provide reference for "Studies have shown that ubiquitin-specific peptidase 47 (USP47) is involved in the occurrence of intestinal injury in sepsis by regulating the inflammatory signaling pathway in intestinal epithelial cells" (line 290).

116) Please change "response[50]" to "response [50]" (line 295).

117) Please replace "the modulation" with "modulation" (line 300).

118) Please format "Informed Consent Statement" using bold (line 315).

Author Response

Comments 1:First of all, we would like to express our gratitude to the editor and reviewers for their constructive comments. The reviewers' questions are presented in bold, and our responses are in regular font. Please refer to the attached file for the detailed content.

Reviewer 2 Report

Comments and Suggestions for Authors

Interesting review about a hot topic.

  1. Abstract: The summary should include the authors' results and conclusions after the review, not just the objectives.
  2. Introduction: A brief description of the review and the aim would be beneficial for readers, providing context and helping them better understand the scope and purpose of the work, (I suggest moving it from the abstract, and reformulate the abstract to be a short description of the review).
  3. There are numerous abbreviations used throughout the text; including a list of abbreviations may be helpful for reader clarity.
  4. Some abbreviations are not described (WWP1, IRAK, IKK, NEMO, etc.), others are not described at first occurrence (TRAF6). Some abbreviations seem to be used differently across the paper ex TNFR and TNFRs, or they are not explained accurately at first occurrence.
  5. I recommend a thorough review of the bibliographic references, ensuring that the original sources are accurately cited.
  6. Some of the bibliographic references appear to be inaccurately attributed and may benefit from a careful review to ensure proper citation. (ex Ref 1, sepsis 3 definition wasn’t established by Wang et colab., but appeard in 2016 in Jama - Singer M, et al. The Third International Consensus Definitions for Sepsis and Septic Shock (Sepsis-3). JAMA. 2016 Feb 23;315(8):801-10).
  7. Same: “According to 27 the global epidemiological data of The Lancet in 2020, approximately 50 million people 28 worldwide suffer from sepsis each year[2, 3]”, but the reference isn’t about article from Lancet.
  8. Line 64 refers to Figure 1a, but there is no 1a, just Figure 1
  9. Regarding Figure 1, is there a potential copyright issue?

Author Response

(The authors gave the same response as above.)

Reviewer 3 Report

Comments and Suggestions for Authors

This manuscript is well-written. The manuscript offers a comprehensive review of the role of ubiquitination in regulating the NLRP3 inflammasome activation and macrophage polarization within the context of sepsis. It delves into the intricate mechanisms by which specific E3 ubiquitin ligases and deubiquitinases modulate inflammatory responses, highlighting their potential as therapeutic targets. I have several comments.

Figure 1 is very good. However, more figure is needed for clear understanding. This manuscript occasionally presents complex mechanisms without sufficient explanatory context, which may hinder comprehension for readers less familiar with the subject matter. Introduce brief overviews or diagrams summarizing key pathways to aid understanding.

Terms such as “non-canonical ubiquitination” and “non-lysine-dependent ubiquitination” are used without clear definitions. Please provide definitions or explanations for specialized terms upon first use to ensure clarity.

Some statements lack appropriate citations. Please check it throughout the manuscript.

While the section on macrophage introduces important concepts, it could benefit from the explanation of neutrophil. Why you do not state in the neutrophil in this manuscript? Maybe, it is also related in some function.

The discussion on HUWE1’s role in promoting K27-linked polyubiquitination of NLRP3 is insightful. However, clarifying how this modification influences NLRP3’s conformational changes would enhance understanding.

The interaction between Praja2 and MFHAS1 in driving M1 polarization is well-presented. Including information on how this interaction affects downstream cytokine production could provide a more comprehensive view.

Comments on the Quality of English Language

None

Author Response

(The authors gave the same response as above.)

Round 2

Reviewer 1 Report

Comments and Suggestions for Authors

Major point:

Please sketch at least one more figure that would summarily illustrate either the macrophage polarization (chapter 4.2) and/or the organ sepsis protection (chapter 5) concept.

Minor points:

1) Please justify all the main text and figure legends.

2) Please remove bold formatting from "Sepsis" (line 9).

3) Please change "cells, and" to "cells and" (line 13).

4) Please replace "the non-traditional" with something like "unconventional" (line 14).

5) Please change "spesis" to "Sepsis", "infection" to "Infection", "organ" to "Organ", "ubiquitination/deu" to "ubiquitination/deu-", "fibrosis." to "fibrosis", "NR F2" to "NRF2", "injury." to "injury", "ph" to "ph-", "ren" to "ren-" in the graphical abstract.

6) The caption "Lung" seems to clash with the top line of the orange box in the graphical abstract. Please fix.

7) Please align the graphical abstract, Figure 1 horizontally so that it becomes centered within the surrounding text.

8) Please replace "Sepsis" with "sepsis" (line 24).

9) Please change "Inflammasome; Septic Organ Injury; Macrophage Polarization; Cardiomyopathy" to "inflammasome; septic organ injury; macrophage polarization; cardiomyopathy" (line 24).

10) Please replace "20-30%" with "20–30%" (line 32).

11) Please change "inflammation (such as the TNF-α and IL-6 storms)" to "inflammation, such as the TNF-α and IL-6 storms," (line 34).

12) Please define abbreviation for "TNF-α" (line 35), "IL-6" (line 35), "RIPK1" (line 53), "USP5" (line 54), "TRIM27" (line 56), "WWP1" (line 77), "cIAPs" (line 104), "IκB-α" (line 106), "Ubc13" (line 125), "TAK1" (line 127), "IKK" (line 129), "NEMO" (line 129), "VANGL2 (line 140), "PDLIM2" (line 141), "GSDMD" (line 165), "ASC" (line 180), "LRR-PYD" (line 184), "AIM2" and "NLRC4" in "AIM2/NLRC4" (line 186), "NEK7" (line 240), "OTU" (line 243), "UBE2M" (line 246), CTLA-4" (line 257), "TRIM27" (line 273), "RNF146" (line 301) and incorporate these into Table 2 as well.

13) Please replace "immunosuppression (such as T cell exhaustion)" with "immunosuppression, such as T cell exhaustion" (line 35).

14) Please change "The organ" to "Organ" (line 36).

15) Please replace "13.8-40%" with "13.8–40%" (line 40).

16) Please change "70-90%" to "70–90%" (line 41).

17) Please replace "post-translational modification" with "post-translational modification (PTM)" (line 43), "post-translational modification" with "PTM" (lines 68, 84), and "post-translational modifications (PTMs)" with "PTMs" (line 316).

18) It is not clear why the authors mention "linear chain" in "Protein ubiquitination, through the E1-E2-E3 enzyme cascade reaction, attaches ubiquitin molecules (76 amino acids) to target proteins via isopeptide bonds (lysine residues) or unconventional bonds (N-terminus, cysteine, etc.), forming various chain types such as K48 (for proteasomal degradation), K63 (for signal activation), and M1 (linear chain, for inflammatory regulation)" (line 45)? Please corroborate in the text.

19) "Protein ubiquitination, through the E1-E2-E3 enzyme cascade reaction, attaches ubiquitin molecules (76 amino acids) to target proteins via isopeptide bonds (lysine residues) or unconventional bonds (N-terminus, cysteine, etc.), forming various chain types such as K48 (for proteasomal degradation), K63 (for signal activation), and M1 (linear chain, for inflammatory regulation)" (line 45) is too long. Please split into at least two sentences.

20) Please replace "ubiquitination, through the E1-E2-E3 enzyme cascade reaction, attaches" with "ubiquitination through the E1-E2-E3 enzyme cascade reaction attaches" (line 45).

21) Please change "cysteine, etc." to "cysteine" (line 47).

22) Please replace "K48 (for proteasomal degradation), K63 (for signal activation), and M1 (linear chain, for inflammatory regulation)" with "K48 for proteasomal degradation, K63 for signal activation, and M1 linear chain, for inflammatory regulation" (line 48).

23) Please specify "it" in "Different from the traditionally recognized "proteasomal degradation function", recent studies have revealed that in sepsis, it regulates the inflammatory signaling pathway, immune cell polarization, and organ protection mechanisms through non-degradative modifications (such as K63/M1 chains)" (line 49).

24) Please change ""proteasomal degradation function"" to "proteasomal degradation function" (line 50).

25) Please replace "as" with "as the" (line 52).

26) Please change "ligase" to "chain assembly" (line 53).

27) Please replace "LUBAC" with "(LUBAC)" (line 53).

28) Please change "the activity of RIPK1" to "its activity" (line 55).

29) Please replace ""ubiquitination axis"" with "ubiquitination axis" (line 58).

30) Please change "three key dimensions, inflammatory" to "inflammatory" (line 61).

31) Please replace "e.g., NF-κB" with "NF-κB" (line 61).

32) Please change "e.g., macrophage" to "macrophage" (line 62).

33) Please replace "e.g., liver" with "liver" (line 63).

34) Please change "one or usually" to "one, or more usually," (line 68).

35) Please replace "of small proteins, ubiquitin chains composed of" with "of" (line 68).

36) Please change "acids, are" to "acids are" (line 69).

37) Please replace "to the" with "to" (line 69).

38) Please provide reference for "The RING-type (such as TRAF6) directly catalyzes the transfer of ubiquitin by recruiting E2 enzymes" (line 75).

39) Please change "TRAF6" to "tumor necrosis factor receptor-associated factor 6 (TRAF6)" (line 76) and "tumor necrosis factor receptor-associated factor 6 (TRAF6)" to "TRAF6" (lines 281, 354).

40) Please provide reference for "The HECT type (such as WWP1) functions by forming a ubiquitin-E3 intermediate" (line 77).

41) Please provide reference for "The RBR type has the characteristics of both" (line 78).

42) Please change "as" to "as in" (line 80).

43) Please replace the straight arrow pointing from E1 to E1-Ub in the activation phase of ubiquitination with a round one so that the E1 substrate and the E1-Ub product become clearly connected in Figure 1.

44) Please replace "ppi" with "PPi", "activation" with "Activation", "conjugation" with "Conjugation", "ligation" with "Ligation" in Figure 1.

45) Please align the "activation", "conjugation", "ligation" captions vertically so that they become equally positioned in Figure 1.

46) Please align "AMP" and "ATP" signs horizontally so that they become centered within the boundaries of their pink boxes in Figure 1.

47) Please align "E1" signs horizontally so that they become centered within the boundaries of their violet boxes in Figure 1.

48) Please align "E2" and "E3" signs horizontally so that they become centered within the boundaries of their cyan boxes in Figure 1.

49) Please replace the three DUBs arrows with only single one in Figure 1.

50) Please describe the depicted ubiquitination mechanism briefly in the legend to Figure 1.

51) Please define abbreviation for "USPs", "UCHs", "OTUs" in the legend to Figure 1.

52) Please change "AMP:" to "AMP," (line 86).

53) Please replace "Adenosine" with "adenosine" (lines 86, 87).

54) Please change "ATP:" to "ATP," (line 86).

55) Please replace "DUB: Deubiquitinase; E1: Ubiquitin-activating enzyme; E2: Ubiquitin-conjugating enzyme; E3: Ubiquitin ligase; K: Lysine; M: Methionine; PPi: Inorganic pyrophosphate; TGF-β: Transforming growth factor-β. Ub: Ubiquitin;" with "DUB, deubiquitinase; E1, ubiquitin-activating enzyme; E2, ubiquitin-conjugating enzyme; E3, ubiquitin ligase; K, lysine; M, methionine; PPi, inorganic pyrophosphate; TGF-β, transforming growth factor-β; Ub, ubiquitin." (line 87).

56) Please provide reference for "With the in-depth study, the non-canonical ubiquitination pathway has received much attention" (line 91).

57) Please change "With the in-depth study" to something like "In addition" (line 91).

58) Please replace "e.g., N-terminal" with "N-terminal" (lines 92, 175).

59) Please change "cysteine" to "and cysteine" (line 93).

60) Please replace "e.g., K27" with "K27" (line 93).

61) Please specify "the traditional pattern" mentioned in "It breaks through the traditional pattern, and unconventional types such as N-terminal formylation have emerged, playing an important role in physiological and pathological processes, especially in inflammation" (line 95).

62) The structure of "It breaks through the traditional pattern, and unconventional types such as N-terminal formylation have emerged, playing an important role in physiological and pathological processes, especially in inflammation" (line 95) seems to be slightly complex. Please simplify this sentence and/or split it in two.

63) Please change "through" to "with" (line 95).

64) Please replace "types" with something like "types of post-translational modifications" (line 95).

65) Please change "NLRP3" to "NOD-like receptor family pyrin domain-containing protein 3 (NLRP3)" and "the NOD-like receptor family pyrin domain-containing protein 3 (NLRP3)" to "NLRP3" (line 154).

66) Please change "non-lysine K27" to "non-K27" (line 99).

67) Please replace "The" with something like "For instance, the" (line 100).

68) Please change "such as M1, K11 linkage, etc." to "M1, K11 linkage" (line 102).

69) Please provide reference for "In the initial stage, cIAPs and RIPK1 are ubiquitinated, and then related factors are recruited" (line 103).

70) Please specify examples of the "related factors" mentioned in "In the initial stage, cIAPs and RIPK1 are ubiquitinated, and then related factors are recruited" (line 103).

71) From "The catalysis of ubiquitin esterification by HOIL-1 is crucial" (line 104) is not clear for what purpose is "catalysis of ubiquitin esterification by HOIL-1" crucial"?

72) Please specify "Downstream" mentioned in "Downstream, the phosphorylation of IκB-α leads to its K48-linked ubiquitination and degradation, promoting the nuclear translocation of NF-κB" (line 105). Downstream of what?

73) Please replace "concept" with something like "repertoire" (line 109).

74) Please provide reference for "For example, in Parkin-mediated mitophagy, damaged mitochondria can recruit the kinase Pink1 and the E3 ligase Parkin" (line 110).

75) Please change "induces the" to "induces" (line 112).

76) Please replace "the physiological and pathological processes of cells" with something like "cell physiology and pathology" (line 114).

77) Please change "The abnormal" to "Abnormal" (line 118).

78) Please replace "Toll-like" with "toll-like" (line 121).

79) Please change "TNFRs (Figure 2), ubiquitination activates the NF-κB signaling pathway through the non-degradative effects of K63/M1-type polyubiquitin chains" to "TNFRs, ubiquitination activates the NF-κB signaling pathway through non-degradative effects of K63/M1-type polyubiquitin chains (Figure 2)" (line 122).

80) Please replace "recruited, which then" with "recruited to bind to" (line 124).

81) Please change "(such as TAK1)" to "such as TAK1" (line 127).

82) From "TNFR stimulation depends on the linear ubiquitin chain assembly complex (LUBAC) to extend the K63 chain with M1-type ubiquitin chains, recruiting the IKK complex through NEMO to release the NF-κB transcription factor and driving the transcription of inflammatory cytokines such as TNF-α and IL-1β (line 127) is not clear from where is "the NF-κB transcription factor" released from? Please mention in the text.

83) Please replace "on the linear ubiquitin chain assembly complex (LUBAC)" with "LUBAC" (line 128).

84) Please change "driving" to "drive" (line 130).

85) Please replace "Illustrates the" with "The" (line 147).

86) "The role of ubiquitination in TLR-like receptors and their downstream inflammatory pathways during sepsis; The function of TNFR-like receptors and their downstream inflammatory pathways in sepsis" (line 148) lacks a verb. Please correct.

87) Please change "sepsis; The" to "sepsis. The" (line 149).

88) Please briefly describe the signal transduction depicted in Figure 2 in its figure legend.

89) Please replace "sepsis, and" with "sepsis and" (line 153).

90) Please change "inflammasome, and" to "inflammasome and" (line 156).

91) Please remove the gap between lines 162 and 163.

92) Please remove the gap between lines 168 and 169.

93) Please provide reference for "HUWE1, a HECT-type E3 ligase, shows specific regulatory functions. In an LPS- and ATP-induced mouse bone marrow-derived macrophage (BMDM) model, it directly interacts with the NACHT domain of NLRP3 via its BH3 domain, triggering non-lysine-dependent K27-linked polyubiquitination modification—a non-canonical modification often misclassified as "non-lysine-dependent"" (line 169).

94) "HUWE1, a HECT-type E3 ligase, shows specific regulatory functions. In an LPS- and ATP-induced mouse bone marrow-derived macrophage (BMDM) model, it directly interacts with the NACHT domain of NLRP3 via its BH3 domain, triggering non-lysine-dependent K27-linked polyubiquitination modification—a non-canonical modification often misclassified as "non-lysine-dependent"" (line 169) is way too long. Please split into at least two sentences.

95) Please remove the gap between lines 178 and 179.

96) Please replace "show" with "show that" (line 179).

97) Please change "NLRP3’s NACHT domain" to "the NACHT domain of NLRP3" (line 179).

98) From "Huwe1-deficient BMDMs exhibit reduced caspase-1 maturation and IL-1β secretion, while BI 8622 inhibition suppresses NLRP3 activation in mouse and human cells" (line 181) is not explicitly clear what target does "BI 8622" inhibit?

99) Please provide reference for "Huwe1-deficient BMDMs exhibit reduced caspase-1 maturation and IL-1β secretion, while BI 8622 inhibition suppresses NLRP3 activation in mouse and human cells" (line 181).

100) Please replace "secretion, while" with "secretion while" (line 182).

101) Please change "BI 8622 inhibition" to "inhibition with BI8622" (line 182).

102) Please replace "NLRP3’s autoinhibitory LRR-PYD interaction" with "the autoinhibitory interaction of NLRP3 with LRR-PYD" (line 184).

103) Please change "HUWE1’s conserved role" to "The conserved role of HUWE1" (line 186).

104) Please replace "unified" with "unifying" (line 187).

105) Please remove the gap between lines 188 and 189.

106) Please change "ubiquitin-specific" to "Ubiquitin-specific" (line 189).

107) Please replace "Lys118" with "K118" (line 192).

108) Please change "will significantly exacerbate" to "significantly exacerbates" (line 193).

109) Please replace "targeting the" with "targeting" (line 195).

110) Please change "and the" to "and" (line 197).

111) Please widen the "Pathological Process", "Key Ubiquitination Event", "Enzyme/Complex Involved", "Functional Impact in Sepsis", and "Ref." columns in Table 1 so that its text becomes less compressed.

112) Please replace "Response" with "response" (Inflammatory Response), "Cell Functions" with "cell functions" (Immune Cell Functions), "Protection" with "protection" (Organ Protection), "recruits" with "recruits the" (NF-κB nondegradative activation), "degradation." with "degradation"  (NF-κB nondegradative activation), "Enhanced" with "Enhances" (NF-κB nondegradative activation), "inflammation." with "inflammation" (NF-κB nondegradative activation, Intestinal barrier protection 2x), "autophagy." with "autophagy" (NLRP3 inflammasome assembly), "Promoted" with "Promotes" (NLRP3 inflammasome assembly), "(USP22)." with "(USP22)" (NLRP3 inflammasome assembly), "Neutrophils" with "neutrophils" (Activation of Neutrophils), "Promotion" with "promotion" (Activation of Neutrophils), "sepsis." with "sepsis" (Activation of Neutrophils), "Cell Functions" with "cell functions" (Immune Cell Functions), "NLRP3." with "NLRP3" (Immune Cell Functions), "(A20)." with "(A20)" (Immune Cell Functions), "activation." with "activation" (T-cell activation inhibition), "immunopathology." with "immunopathology" (T-cell activation inhibition), "signaling." with "signaling" (Lung injury regulation), "(TRAF6/USP38)." with "(TRAF6/USP38)" (Lung injury regulation), "ubiquitination." with "ubiquitination" (Liver anti-oxidative stress), "(RNF146)." with "(RNF146)" (Liver anti-oxidative stress), "expression." with "expression" (Cardiomyocyte pyroptosis), "dysfunction." with "dysfunction" (Cardiomyocyte pyroptosis), "injury." with "injury" (Renal mitophagy activation), "clearance." with "clearance" (Renal mitophagy activation) in Table 1.

113) Please remove italics formatting from "Neutrophils, acting as the primary immunological defense in sepsis, experience complex functional control by the ubiquitination network. Heparin-binding protein (HBP) released during sepsis obstructs K48-linked ubiquitination of the E3 ubiquitin ligase TRIM21, thereby stabilizing TRIM21 and facilitating K63-linked ubiquitination of transcription factor P65. This process mediates pulmonary microvascular endothelial hyperpermeability and glycolytic dysfunction through the TRIM21-P65 signaling axis, thus contributing to the pathogenesis of acute lung injury (ALI) [54]. Mechanistically, heat shock protein Hsp90 modulates neutrophil apoptosis by maintaining the c-Src/caspase-8 complex, hence inhibiting its ubiquitination [55]. The E3 ligase Mid1 facili-tates neutrophil-endothelial adhesion by downregulating PP2Ac, hence promoting 214 ICAM-1 expression, whereas silencing Mid1 mitigates septic lung injury by inhibiting the Mid1-PP2Ac axis [56]. These data collectively indicate that ubiquitination alterations play a crucial role in sepsis-induced acute lung injury by modulating neutrophil activation, apoptosis, and endothelial interactions, underscoring the ubiquitination network as a potential therapeutic target for sepsis-related lung injury." (line 206).

114) Please provide reference for "Heparin-binding protein (HBP) released during sepsis obstructs K48-linked ubiquitination of the E3 ubiquitin ligase TRIM21, thereby stabilizing TRIM21 and facilitating K63-linked ubiquitination of transcription factor P65" (line 207).

115) Please change "acute lung injury" to "ALI" (line 217).

116) Please provide reference for "The E3 ubiquitin ligase Praja2 can bind to MFHAS1 and mediate its non-degradative ubiquitination, promoting MFHAS1 accumulation" (line 228).

117) Please replace "to" with "to the" (line 231).

118) Please change "affects M1 markers exclusively" to "exclusively affects M1 markers" (line 235).

119) Please replace "[62], and" with "[62]. And" (line 236).

120) Please format "+" in "Ly6C+" using superscript (line 237).

121) Please change "ubiquitination (with key functional sites at K189 and K293 residues of NEK7)" to "ubiquitination with key functional sites at K189 and K293 residues of NEK7" (line 241).

122) Please replace "ZnF 4/ZnF 7" with "ZnF4/ZnF7" (line 243).

123) Please remove the gap between lines 245 and 246.

124) Please change "e.g., IL-1β" to "IL-1β" (line 248).

125) Please replace "TNF-α" with "and TNF-α" (line 249).

126) Please change "clearance ability" to "clearance" (line 249).

127) Please replace "the deletion" with "deletion" (line 250).

128) Please change "sepsis, and" to "sepsis and" (line 254).

129) From "Among them, casitas B lymphoma-b (Cbl-b) is a key downstream regulator of the CD28 and CTLA-4 co-stimulation/co-inhibition signaling pathway, and E3 ubiquitin ligases play a central role in the regulation of effector T cell functions" (line 256) is not unequivocally clear whether the authors mean to say "CTLA-4 co-stimulation and co-inhibition signaling pathway" or "CTLA-4 co-stimulation or co-inhibition signaling pathway"?

130) Please replace "(such as binding to TCR signaling molecules such as LCK, SLP76, and ZAP70)" with "such as binding to TCR signaling molecules such as LCK, SLP76, and ZAP70" (line 259).

131) Please change "Lys33" to "K33" (line 261).

132) Please replace "kinase, and this" with "kinase. This" (line 264).

133) Please change "cells (independent of CD28 stimulation)" to "cells, independent of CD28 stimulation" (line 265).

134) Please replace "immunotherapy," with something like "immunotherapy as" or "immunotherapy because" (line 267).

135) Please provide reference for "Knocking down TRIM27 can inhibit the ubiquitination degradation of peroxisome proliferator-activated receptor γ (PPARγ), reduce the expression of NADPH oxidase 4 (NOX4) and the activation of the downstream p-p65 inflammatory pathway, alleviate the inflammatory infiltration, apoptosis, and oxidative stress injury of lung tissues" (line 274).

136) Please change ""TRIM27-PPARγ-NOX4"" to "TRIM27-PPARγ-NOX4" (line 279).

137) Please replace "IL-33R (interleukin 33 receptor)" with "interleukin 33 receptor (IL-33R)" (line 283).

138) Please remove the gap between lines 298 and 299.

139) Please change "poly (ADP-ribose)" to "poly(ADP-ribose)" (line 299).

140) Please replace ""RNF146/PARP1 ubiquitination axis"" with "RNF146/PARP1 ubiquitination axis" (line 310).

141) Please change "deubiquitinase (DUB)" to "DUB" (line 323).

142) Please replace "ubiquitin-specific peptidase 7 (USP7)" with "USP7" (line 324).

143) Please remove the gap between lines 338 and 339.

144) Please change "ubiquitin-specific peptidase 47 (USP47)" to "USP47" (line 351).

145) Please format "Table 2. List of Abbreviations" (line 371) consistently with "Table 1. Non-Traditional Roles of Protein Ubiquitination in Sepsis Pathogenesis" (line 202).

146) Please widen the "Abbreviation" columns in Table 2 so that its text becomes less compressed (2x).

147) Please replace "Term" with "term" (2x) in Table 2.

148) Table 2 seems to clash with the Acknowledgements section. Please resolve this issue.

Author Response

The Editors

Cells

Dear Editors,

We would like to take the opportunity to thank you for allowing us to resubmit our manuscript (The Role of Protein Ubiquitination in the Onset and Progression of Sepsis, Manuscript ID: cells-3660493). We would also like to thank the reviewers for their insightful comments. As you will see, we have carefully considered all the reviewers’ comments and revised the manuscript accordingly. We also added a figure to the revised manuscript to better highlight its key concepts.

Attached please find our revised manuscript and a point-by-point answer to the editor and reviewers’ comments. We hope that you and the reviewers will find that with these changes our manuscript is now suitable for publication in Cells. Thank you!

Sincerely,

Min Fang, Ph.D, Professor

Reviewer 2 Report

Comments and Suggestions for Authors

Thank you for addressing the suggested modifications.

Author Response

We appreciate your thorough review and constructive feedback, which have been instrumental in improving the manuscript.

Round 3

Reviewer 1 Report

Comments and Suggestions for Authors

Major point:

Some of the entries presented in Table 2 seem to be cut through. Please fix.

Minor points:

1) Please change "20 - 30%" to "20–30%" (line 32).

2) Please replace "13.8 - 40%" with "13.8–40%" (line 41).

3) Please change "70 - 90%" to "70–90%" (line 41).

4) Please replace "modification(PTM)" with "modification (PTM)" (line 44).

5) Please change "N-terminus, or" to "N-terminus or" (line 48).

6) Please replace "M1 chains (linear chains)" with "linear M1 chains" (line 50).

7) Please change "1(RIPK1)" to "1 (RIPK1)" (line 55).

8) Please replace "(NF-κB non-degradative signaling)" with "such as NF-κB non-degradative signaling" (line 64).

9) Please change "(macrophage M1/M2 polarization)" to "such as macrophage M1/M2 polarization" (line 65).

10) Please replace "(liver anti-oxidative stress pathways)" with "such as liver anti-oxidative stress pathways" (line 65).

11) Please define abbreviation for "RBR" (line 77), "MyD88" (line 137), "IRAK1/4" (line 137), "ATG5" in "ATG5-mediated" (line 216), "PP2Ac" (line 240), "ICAM-1" (line 241), "JNK" in "TLR2/JNK/NF-κB" (line 254), "Praja2" (line 254), "iNOS" (line 260), "Arg-1" (line 260), "MMR" (line 261), "LCK" (line 291), "SLP76" (line 291), "ZAP70" (line 292).

12) Please change "ww domain-containing e3" to "WW domain-containing E3" (line 79).

13) Please change "M1 chains (linear chains)" to "linear M1 chains" (line 83).

14) Please replace "Deubiquitinase" with "deubiquitinase" (line 95).

15) Please change "Ubiquitin-activating" to "ubiquitin-activating" (line 95).

16) Please replace "Ubiquitin-conjugating" with "ubiquitin-conjugating" (line 95).

17) Please change "Ubiquitin" to "ubiquitin" (lines 96, 97).

18) Please replace "Lysine" with "lysine" (line 96).

19) Please change "Methionine" to "methionine" (line 96).

20) Please replace "Ovarian Tumor" with "ovarian tumor" (line 96).

21) Please change "Inorganic" to "inorganic" (line 96).

22) Please replace "Transforming" with "transforming" (line 97).

23) Please change "USPs, Ubiquitin-Specific Proteases; UCHs, Ubiquitin C-Terminal Hydrolases;" to "UCHs, ubiquitin C-terminal hydrolases; USPs, ubiquitin-specific proteases." (line 97).

24) Please provide reference for "This term denotes modifications via non-lysine residues (N-terminal formylation, and cysteine thiolation) or atypical lysine chain linkages (K27, M1), which differ from canonical K48/K63-linked degradation or signaling pathways" (line 101).

25) Please replace "formylation," with "formylation" (line 101).

26) Please change "(TNFR1/TRAF2)" to "such as TNFR1/TRAF2", "such as TNFR1 or TRAF2", or "such as TNFR1 and TRAF2" (line 115).

27) Please replace "Downstream (Ser 32 and 36), the phosphorylation of the inhibitor of nuclear factor kappa-b alpha (IκB-α)" with "Downstream, phosphorylation of the inhibitor of nuclear factor kappa-b alpha (IκB-α) at Ser32 and 36" (line 118).

28) Please change "Pink1" to "PINK1" (line 125).

29) Please replace "Pink1-dependent" with "PINK1-dependent" (line 126).

30) Please change "e2 n" to "E2 N" (line 138).

31) Please provide reference for "TNFR stimulation depends on LUBAC to extend K63-linked ubiquitination on NEMO with M1-type (linear) ubiquitin chains" (line 141).

32) Please replace "NEMO" with "the NF-κB essential modulator (NEMO)" (line 142) and "NF-κB essential modulator (NEMO)" with "NEMO" (line 143).

33) Please change "M1-type (linear)" to "linear M1-type" (lines 142, 154).

34) Please specify the "NF-κB transcription factors" mentioned in "This modification recruits the IκB kinase (IKK) complex through NF-κB essential modulator (NEMO), leading to the release of NF-κB transcription factors from their cytoplasmic inhibitory complex with IκBα, which then drives the transcription of inflammatory cytokines such as TNF-α and IL-1β" (line 143).

35) Please replace "of" with "of the" (line 144).

36) Please change "the linear ubiquitin chain assembly complex (LUBAC)" to "LUBAC" (line 154).

37) Please replace "IκB kinase (IKK)" with "IKK" (line 155).

38) "TNFR stimulation depends on the linear ubiquitin chain assembly complex (LUBAC) to extend K63-linked ubiquitination on NEMO with M1-type (linear) ubiquitin chains. This modification recruits the IκB kinase (IKK) complex through NEMO, leading to the release of NF-κB transcription factors from their cytoplasmic inhibitory complex with IκBα, which then drives the transcription of inflammatory cytokines (TNF-α and IL-1β)" (line 153) seems to be duplicit to "TNFR stimulation depends on LUBAC to extend K63-linked ubiquitination on NEMO with M1-type (linear) ubiquitin chains. This modification recruits the IκB kinase (IKK) complex through NF-κB essential modulator (NEMO), leading to the release of NF-κB transcription factors from their cytoplasmic inhibitory complex with IκBα, which then drives the transcription of inflammatory cytokines such as TNF-α and IL-1β" (line 141). Please remove one of the two instances.

39) Please change "(LPS) induced" to "(LPS)-induced" (line 161).

40) The blue (K48 UB), green (K63 Ub), and red (M1 Ub) circles seem to be cut through in Figure 2. Please correct.

41) Please replace "Gasdermin" with "gasdermin" (line 189).

42) Please provide reference for "HUWE1, a HECT-type E3 ligase, shows specific regulatory functions. In an LPS-and ATP-induced mouse bone marrow-derived macrophage (BMDM) model, it directly interacts with the NACHT domain of NLRP3 via its BH3 domain" (line 193).

43) Please change "card" to "CARD" (lines 205, 212).

44) Please replace "BI 8622" with "BI8622" (line 207).

45) Please change "(AIM2)/nlr" to "(AIM2)/NLR" (line 212).

46) Please replace "exacerbates the" with "exacerbates" (line 219).

47) Please change "P65" to "p65" in Table 1 (Activation of neutrophils).

48) Please replace "P65" with "p65" (line 235).

49) Please change "TRIM21-P65" to "TRIM21-p65" (line 236).

50) Please replace "IL-1β" with "IL-1β," (line 260).

51) Please change "Arg-1" to "Arg-1," (line 260).

52) Please replace "and" with "and the" (line 267).

53) Please remove bold formatting from "Mechanistic model of MFHAS1 ubiquitylation-mediated regulation of the MAPK pathway during macrophage polarization. Ubiquitylation of MFHAS1 by Praja2 activates the TLR2/JNK/p38/NF-κB signaling cascade, driving M1 macrophage polarization and promoting M2-to-M1 phenotypic transition" and "." in "Mechanistic model of MFHAS1 ubiquitylation-mediated regulation of the MAPK pathway during macrophage polarization. Ubiquitylation of MFHAS1 by Praja2 activates the TLR2/JNK/p38/NF-κB signaling cascade, driving M1 macrophage polarization and promoting M2-to-M1 phenotypic transition [43]." (line 280).

54) Please make a gap between lines 387 and 388.

55) Please change "Tumor" to "tumor" in Table 2 (cIAPs).

Author Response

Dear Reviewer#1 Dear Reviewer, We are truly grateful for your meticulous review and the insightful comments you have provided. Your feedback has been invaluable in helping us improve our manuscript. Below are our detailed responses to each of your comments. We have incorporated the revisions as described, and the detailed changes are provided in the attached file for your reference.
